# Prevalence and determinants of depression and/or anxiety among adults using Kenya Demographic and Health Survey of 2022: Multilevel logistic regression analysis

**Mamaru Melkam** [1]*, **Setegn Fentahun**[1], **Girmaw Medfu Takelle**[1], **Gidey Rtbey**[1], **Fantahun Andualem**[1], **Girum Nakie**[1], **Gebresilassie Tadesse**[1], **Yilkal Abebaw Wassie**[2]

**1** Department of Psychiatry, University of Gondar, College of Medicine and Health Science, Gondar, Ethiopia, **2** Department of Medical Nursing, University of Gondar, College of Medicine Health Science, Gondar, Ethiopia

* mamarumelkam@gmail.com

## Abstract

### Introduction

Depression and/or anxiety can be persistent or recurrent significantly affecting a person's capacity to manage daily life, job, and school. The burden of depression and anxiety is rising from time to time, with serious consequences for overall health. Depression and anxiety are crippling conditions that can impact individuals of the whole community. Despite the high prevalence of depression and/or anxiety few studies were conducted that show the diagnosis levels of depression and/or anxiety in the community, particularly in Kenya. Therefore, this study aims to determine the prevalence of depression and/or anxiety and their determinant factors among adults in Kenya using data sourced from the 2022 Kenya Demographic and Health Survey.

### Method

The Kenya demographic and health survey of 2022 data were used for this secondary data analysis in 2024. The survey included age groups ranging from 15 to 49, with a total sample size of 16,901 participants. Multilevel analysis was used to determine the prevalence of depression and/or anxiety with determinant factors at the 95% CI.

### Results

The overall prevalence of depression and/or anxiety was 3.84% with a 95% CI of (3.56, 4.14). Of this, 2.85% have only depression, 1.97% have only anxiety disorders, and 0.98% have comorbid depression and anxiety. In multivariable multilevel logistic regression analysis sexually violated, having a chronic medical illness, being divorced and widowed, having a job, and being HIV positive were associated with depression and/or anxiety with a p-value of less than 0.05.

**Data availability statement:** No.

**Funding:** The author(s) received no specific funding for this work.

**Competing interests:** No competing interest.

**Abbreviation:** AIC; Akaike Information Criteria, AOR: Adjusted Odd Ratio, DHS: Demographic Health Data, CI: Confidence Interval, ICC: Intra-Class Correlation, MOR: Median Odds Ratio, PCV: Proportional Change in Variance, WHO: World Health Organizations.

## Conclusions

According to the findings of this study the prevalence of depression and/or anxiety was 3.84%. This finding poses a significant challenge for the community to perform their daily tasks. As a result, the healthcare systems of Kenya have to mitigate the burden of depression and/or anxiety. All the clients must be treated since they received a diagnosis as reported by the physician.

## Introduction

Depressive disorders are characterized by loss of interest or pleasure, depressed mood, feelings of guilt or low self-worth, disturbed sleep or appetite, feelings of tiredness, suicide, and poor concentration [1]. Anxiety can be defined as feelings of tension, uneasiness, nervousness, fear, and high autonomic activity with varying degrees of intensity and excessive wornness with a catastrophic future [2]. According to the Diagnostic and Statistical Manual-V (DSM-V) anxiety disorders refer to a group of mental disorders characterized by feelings of anxiety and fear, including generalized anxiety disorder, panic disorder, phobias, social anxiety disorder, and anxiety secondary to medical conditions and substance use [2]. Depression and anxiety share common clinical features including; fatigue or loss of energy, difficulty in concentration, psychomotor disturbance, and disturbed sleep or appetite [3].

According to a World Health Organization (WHO) report, depression and anxiety are the most common disorders among other mental illnesses worldwide [4]. They are the most significant health indicators that contribute significantly to morbidity in mental health. A 2008 WHO survey stated that one in five persons had experienced depression and anxiety in the previous year and that 29.2% had encountered mental disease at some point in their lives [5]. Depression and anxiety are the most common mental health disorders in the community due to their devastating impact on serious public health problems with a recent global burden of 4.4%[1,6]. Depression is a prevalent mental health condition that impacts over 300 million individuals globally [1]. In the world in 2013, one in nine individuals suffered from an anxiety disorder [7]. The worldwide projected percentage of the world's population suffering from anxiety disorders in 2015 was 3.6%, which is more common among females than males (4.6% compared to 2.6% at the global level) [1].

According to a recent study, there are huge variations in the prevalence of depression and anxiety from place to place based on the cultural and geographic context [8]. It can also aid in the planning of community-based preventive initiatives and the allocation of public health and clinical treatment resources to populations that most need them [9]. An estimated 300 million individuals worldwide or 4.4% of the global population are thought to suffer from depression. Anxiety and depression frequently prevent people from engaging in daily activities, such as caring for their families or working efficiently [10]. In the China population survey study the burden of comorbid depression and anxiety was 9.05% with 10.6% depression and 12.8% anxiety [11]. The comorbid prevalence of depression and anxiety was 2% in the New York community [12]. In Australia, the prevalence of depression and anxiety at the community level was 10% [13]. The prevalence of depression among adults ranges between 11.4–39.6% [14–18] while the incidence of anxiety is between 14.2–37.7% [18]. The prevalence of depression and anxiety in Malaysia was 12.3% and in Pakistan, a systematic review and meta-analysis on depression and anxiety revealed 34% [17,19]. In Ghana, a community-based study the prevalence of depression and anxiety was 24.2% [20]. The prevalence of depression and anxiety in low and middle-income countries was 2.3% [21]. The prevalence of depression and anxiety in Ethiopia was 10.04% [22] Somalia and Kenya refugees were 33.6% [23].

From the previous studies, there are a lot of factors that were associated with depression and/or anxiety including; female sex, age, marital status, having a diagnosed chronic noncommunicable medical diseases and alcohol consumption, mass media exposure, and low income the most statistically important [19,20,24]. Education level, employment status, living conditions, hypertension, use of cigarettes, physical inactivity, and sexual violence were also other factors that were associated with depression and/or anxiety [18,25–27]. Not being treated and physically inactive/or not doing exercise were also the most determinant significant factors for anxiety and depression [28, 29].

Even though there is a large burden of depression and anxiety at the community level, there are no studies conducted in Kenya at the community level. Depression and anxiety are the most important mental disorders that need great attention and priority to mitigate the magnitude and risk factors, especially in low and middle-income countries including Kenya. To have effective interventions for common mental problems particularly, depression and anxiety this study will give evidence for prevention, intervention, and policy-makers from KDHS (Kenya Demographic and Health Survey). The prevalence of depression and anxiety are conducted in different populations among medical clients, students, and immigrants assessed with screening tools rather than diagnostic levels [22,30,31]. Depression and anxiety are not conducted at the level of diagnostic levels which means this study revealed the participants who have leveled as they have depression and anxiety by physician at the community survey level. With these considerations, this study aims to determine the prevalence and determinant factors of depression and/or anxiety among adults in Kenya based on Demographic and Health Survey data of 2022.

## Methods and materials

### Study design, setting, and participants

Community-based multilevel secondary data analysis was employed in 2024, from the 2022 Kenya Demographic and Health Survey (KDHS). This study was conducted from the Kenya Demographic Health Survey data of 2022 from Kenya. KDHS is the seventh survey conducted in Kenya. A two-stage stratified sampling design was used for the Kenya Demographic and Health Survey (KDHS) in 2022. Using the equal probability selection method, 1,692 clusters were chosen in the first stage from the Kenya household health survey framework. The survey included men, women, children, births, and households' datasets from the KDHS. From this survey, the extracted data was the Individual Record dataset (IR file). Men and women participants between the ages of 15 to 49 were selected from the Kenya community as the source of populations. A total of 16,901 weighted samples were used as a final analysis for this study from the clusters. Detailed information can be accessed from this official dataset link http://www.dhsprogram.com/.

### Measurement variables of the study

**Dependent variables.** The outcome variables were measured by the diagnostic criteria and participants were leveled as having depression and anxiety rather than using screening tools. Told by a doctor they have the diagnosis of depression and anxiety were taken with assessment done by physicians' diagnosis criteria. The outcomes are not assessed by screening material or tools to identify participants who have depression and anxiety rather participants are leveled by doctors as they have depression and anxiety. The diagnosis of the two disorders is generated as a single variable and positive either for depression and/or anxiety were coded 1 and the otherwise 0. Therefore, the dependent variable was used from the KDHS definitions as reported in 2022.

**Independent variables.** Educational level, age, marital status, sex, religion, ethnicity, current use of cigarettes and alcohol, wealth status, occupations, and residence were included as sociodemographic variables. Sexual violence, chronic medical illness (having at least one of the following diseases; DM, hypertension, and heart disease), HIV positive, and physical exercise were considered as the clinical and behavioral variables extracted from the KDHS based on the literature review conducted previously in depression and anxiety. Mass media exposure and residence were considered for community-level variables in this study.

**Data management and analysis.** The extracted Kenya DHS data have sociodemographic, behavioral, and clinical characteristics. The extracted data was cleaned and recorded for further analysis with Stata version 14. The descriptive statistics including frequency and percentage were done in text and table. To maintain the hierarchical nature of the extracted data a mixed multilevel analysis was conducted. Variables with a p-value of less than 0.25 from bivariable multilevel logistic regression were selected to be further analyzed in multivariable analysis and p-value less than 0.05 were statistically significantly associated. Adjusted Odd Ratio (AOR) and 95% Confidence Interval (CI) for the associated variables with depression and/or anxiety were employed.

Four model analyses were contracted for multivariable multilevel logistic regression analysis. The first model was a null model or model one conducted without explanatory variables. The second model fitted the individual-level variables only, the third model contained community-level variables, and the fourth model fitted both individual and community-level variables. Deviance and Akaike Information Criterion (AIC), were used for model comparison and fitness, from this analysis having the lowest score was considered the best-fitted model.

Additionally, the random effect of depression and/or anxiety measure of variation across residence clusters was done by Intra Class Correlation (ICC) and Median Odds Ratio (MOR). The degree of homogeneity of depression and/or anxiety measurement, and the variation of depression and/or anxiety in the cluster by odd ratio scale measurement were performed by ICC and MOR respectively [32]. Finally, the AOR with 95% CI was included and factors with p-values less than 0.05 were considered statistically significantly associated.

**Ethics approval and consent to participate.** This study did not need ethical clearance since we used secondary data without direct contact with the study participant. The data was obtained from the measure of DHS program and we have received permission to access the data online with a request to the measure DHS program http://www.dhsprogram.com. The data is available online for everyone publicly. The ethical approval detail information authorized to download Survey data from the Demographic and Health Surveys (DHS) Program is accepted.

## Results

### Descriptive characteristics of the participant

A total of 16,901 study participants between the ages of 15 to 49 were included in this secondary data analysis. Of the study participants, 14453(85.52%) were male. About 5,657(33.47%) were protestant religious followers and more than half of the study participants 10,384(61.44%) were from rural areas. More than half of the study participants were married and more than half of the study participants 9,158(54.24%) had a job (Table 1).

### Clinical and behavioral characteristics of the study

Of the study participants, 15,388(91.05%) had at least one chronic medical illness, and 400(2.37%) of the study participants were declared HIV positive. From the study participants, 15,403(91.14%) were alcohol users almost every day and 1,378(8.15%) had family/parental

**Table 1. Descriptive characteristics of the study participants by socio-demographic and depression and/or anxiety (n = 16,901).**

| Variables | Categories | Weighted sample (%) | Depression and/or anxiety | |
|---|---|---|---|---|
| | | | Yes | No |
| **Sex** | Male | 14453(85.52) | 554 | 13889 |
| | Female | 2448(14.48) | 95 | 2353 |
| **Age** | 15-19 | 3339(19.76) | 61 | 3276 |
| | 20-29 | 5895(34.88) | 208 | 5687 |
| | 30-39 | 4689(27.74) | 218 | 4471 |
| | 40-49 | 2978(17.62) | 160 | 2818 |
| **Region** | Catholic | 3,005(17.78) | 138 | 2867 |
| | Protestant | 5,657(33.47) | 218 | 5439 |
| | Evangelical Church | 3,581(21.19) | 141 | 3440 |
| | African instituted church | 1,298(7.68) | 50 | 1248 |
| | Islam | 2,637(15.60) | 72 | 2565 |
| | Others religions* | 723(4.27) | 30 | 693 |
| **Mass media exposure** | Yes | 14253(84.32) | 354 | 14 607 |
| | No | 2847(14.48) | 96 | 2943 |
| **Educations** | No Education | 2,075(16.84) | 62 | 2013 |
| | Primary | 6,171(36.51) | 253 | 5918 |
| | Secondary | 6,067(35.90) | 200 | 5867 |
| | Higher | 2,588(15.31) | 134 | 2454 |
| **Residence** | Urban | 6,517(38.56) | 298 | 6219 |
| | Rural | 10,384(61.44) | 351 | 10033 |
| **Ethnicity** | Kalenjin | 3,339(19.76) | 106 | 3233 |
| | Kamba | 1,457(8.62) | 31 | 1426 |
| | Kikuyu | 2,326(13.76) | 105 | 2221 |
| | Luhya | 2,178(12.89) | 99 | 2079 |
| | Luo | 2,557(15.13) | 86 | 2471 |
| | Meru | 1,105(6.54) | 50 | 1055 |
| | Somali | 1,326(7.85) | 25 | 1301 |
| | Others ethnicity ** | 2613(15.46) | 178 | 2416 |
| **Occupations** | No job | 7,725(45.76) | 179 | 7546 |
| | Have job | 9,158(54.24) | 469 | 8689 |
| **Marital status** | Never in union | 5,259(31.12) | 132 | 5127 |
| | Married | 8,685(51.39) | 325 | 8360 |
| | Widowed/separated | 2,957(17.50) | 192 | 2765 |
| **Wealth index** | Poorest | 3,758(22.24) | 113 | 3615 |
| | Poorer | 2,975(17.60) | 104 | 2871 |
| | Middle | 3,308(19.57) | 133 | 3175 |
| | Richer | 3,753(22.21) | 145 | 3608 |
| | Richest | 3,107(18.38) | 154 | 2953 |

*Other religions (Hindu, orthodox, atheist, traditionist).

**Other ethnicity (embu, kisii, maasai, mijikenda/Swahili, taita/taveta).

alcohol conceptions. Almost all the study participants 16,795(99.37%) use cigarettes despite 12,418(73.47%) doing physical exercise. Of the study participants, 488(2.89%) faced sexual violations based on their gender (Table 2).

## Prevalence of depression and/or anxiety

The overall prevalence of depression and/or anxiety from the KDHS data was 3.84% with a 95% CI of (3.56,4.14). Of this, 2.85% have only depression, 1.97% have only anxiety disorders, and 0.98% have comorbid depression and anxiety. From the overall prevalence 3.85% of depression and/or anxiety 85% were male participants and 15% were female participants.

**Table 2. Clinical and behavioral characteristics of the study participants (n = 16901).**

| Variables | Categories | Weighted sample (%) | Depression and/or anxiety | |
|---|---|---|---|---|
| | | | Yes | No |
| **Chronic medical illness** | Yes | 15,388(91.05) | 471 | 14,917 |
| | No | 1,513(8.95) | 178 | 1,335 |
| **HIV positive** | Yes | 400(2.37) | 35 | 363 |
| | No | 16,501(97.63) | 612 | 15,889 |
| **Alcohol use** | Not at all | 818(4.84) | 55 | 763 |
| | 1-5 days | 537(3.18) | 55 | 482 |
| | 6-10 days | 89(0.53) | 9 | 80 |
| | 11-24 days | 54(0.32) | 6 | 48 |
| | Almost every day | 15,403(91.14) | 524 | 14,879 |
| **Family alcohol use** | Yes | 1,378(8.15) | 104 | 1274 |
| | No | 15,523(91.85) | 545 | 14,978 |
| **Cigarette use** | Yes | 16,795(99.37) | 639 | 16,156 |
| | No | 106(0.63) | 10 | 96 |
| **Physical exercise** | Active | 12,418(73.47) | 488 | 11,930 |
| | Inactive | 4,483(26.53) | 161 | 4322 |
| **Sexual violence** | Violated | 488(2.89) | 62 | 426 |
| | Not violated | 16,413(97.11) | 582 | 15,826 |

## Model fitness and statistical analysis

The ICC in the null model (model one) was a 21.21% variation of the participants who have depression and/or anxiety related to the attributed to the cluster. The null model's MOR of depression and/or anxiety was 1.85, suggesting that there was variation amongst the clusters. The odds of a single participant with depression and/or anxiety were 1.85 times higher in the cluster with a higher risk of these conditions than in the cluster with a lower risk, if that person was chosen at random from each of the two clusters. The lowest deviation value was used to select the best fitting; therefore, model IV was the best model for this study (Table 3).

## Associated factors with depression and/or anxiety

In bivariable logistic regression analysis age, marital status, alcohol use, occupation, cigarette use, educational level, HIV positive, having chronic medical illness, sexual violence, family member alcohol use, physically inactive were associated with depression and/or anxiety with p value less than 0.25. In multivariable multilevel logistic regression analysis sexually violated, having a chronic medical illness, being divorced and widowed, having a job, and HIV positive were associated with depression and/or anxiety with a p-value of less than 0.05. The development of depression and/or anxiety was 1.39 times more likely among divorced and widowed [AOR = 1.39; 95% CI: (1.14, 1.69)]. The odds of depression and/or anxiety development were 2.09 times higher among HIV-positive subjects than negative subjects [AOR = 2.09; 95% CI: (1.45, 3.02)]. Being sexually violated was 2.81 times higher than not experiencing sexual violence to have depression and/or anxiety [AOR = 2.81;95% CI: (2.05, 3.85)]. The odds of depression and/or anxiety development were 1.55 times higher among participants who have a job as compared with participants who have been between jobs [AOR = 1.55; 95% CI: (1.27, 1.90)]. The odds of experiencing depression and/or anxiety were 3.50 times higher among participants who have chronic medical illnesses compared to the others who haven't chronic medical illnesses [AOR = 3.50; 95% CI: (2.89, 4.23)] (Table 3).

**Table 3. Multilevel analysis of variables associated with depression and/or anxiety among DHS of Kenya, 2022.**

| Variables | Null model | Model I | Model II | Model III |
|---|---|---|---|---|
| **Age** | | | | |
| 15-19 | | 1.00 | | 1.00 |
| 20-29 | | 1.08(0.77,1.51) | | 1.07(0.76, 1.50) |
| 30-39 | | 1.18(0.81, 1.71) | | 1.17(0.80, 1.70) |
| 40-49 | | 1.13(0.77, 1.67) | | 1.13(0.76, 1.67) |
| **Cigarette use** | | | | |
| No | | 1.00 | | 1.00 |
| yes | | 1.80(0.87, 3.71) | | 1.79(0.87, 3.71) |
| **HIV Positive** | | | | |
| No | | 1.00 | | 1.00 |
| Yes | | **2.09(1.45, 3.02)*** | | 2.10(1.45, 3.03) |
| **Education** | | | | |
| No education | | 0.87(0.61, 1.22) | | 0.89(0.63, 1.25) |
| Primary | | 0.81(0.63, 1.03) | | 0.83(0.65, 1.05) |
| Secondary | | 0.80(0.63, 1.03) | | 0.82(0.64, 104) |
| Higher | | 1.00 | | 1.00 |
| **Marital status** | | | | |
| Married | | 1.00 | | 1.00 |
| Single | | 0.93(0.72, 1.21) | | 0.93(0.71, 1.20) |
| Divorce/widowed | | **1.39(1.14, 1.69)*** | | **1.38(1.13, 1.67)** |
| **Occupation** | | | | |
| Haven't job | | 1.00 | | 1.00 |
| Have job | | **1.55(1.27, 1.90)*** | | 1.55(1.26, 1.89) |
| **Sexual violence** | | | | |
| Not violated | | 1.00 | | 1.00 |
| Being violated | | **2.81(2.05, 3.85)*** | | 2.85(2.05, 3.86) |
| **Family substance use** | | | | |
| No | | 1.00 | | 1.00 |
| Yes | | 1.20(0.94, 1.54) | | 1.20(0.94, 1.54) |
| **Chronic medical illness** | | | | |
| No | | 1.00 | | 1.00 |
| Yes | | **3.50(2.89, 4.23)*** | | 3.49(2.88, 4.22) |
| **Physical exercise** | | | | |
| Inactive | | 1.00 | | 1.00 |
| Active | | 0.99(0.82, 1.20) | | 0.99(0.81, 1.19) |
| **Alcohol use** | | | | |
| Not drink | | 1.00 | | 1.00 |
| Drink 1-5 days | | 1.60(0.07, 2.39) | | 1.59(0.06, 2.38) |
| Drink 6-10 days | | 1.46(0.68, 3.17) | | 1.46(0.67, 3.16) |
| Drink 11-24 days | | 1.54(0.57, 4.15) | | 1.52(0.56, 4.12) |
| Drink almost every day | | 0.70(0.52, 4.95) | | 0.71(0.52, 4.95) |
| **Community level analysis** | | | | |
| **Residence** | | | | |
| Rural | | | 1.00 | 1.00 |
| Urban | | | 1.37(1.17, 1.60) | 1.17(0.99, 1.39) |
| Mass media exposure | | | 1.48(1.20, 1.72) | 1.51(0.88, 2.01) |
| High | | | 1 | 1 |
| Low | | | | |

*(Continued)*

**Table 3.** (Continued)

| Variables | Null model | Model I | Model II | Model III |
|---|---|---|---|---|
| **Model fit statistics** | **Model I** | **Model II** | **Model III** | **Model IV** |
| Log likely ratio test | -2767.27 | -2568.14 | -2764.79 | -2566.68 |
| Deviance | 5496.0775 | 5133.3654 | 5529.5805 | 5096.7952 |
| AIC | 5179.318 | 5178.283 | 5535.581 | 5177.365 |
| BIC | 5437.946 | 5340.698 | 5558.786 | 5337.515 |
| ICC | .0212171 | | | |
| MOR | 1.85 | | | |

*ICC: Intra-Class Correlation.

*MOR: Median Odds Ratio.

*AIC: Akaike Information Criterion.

*BIC: Bayesian Information Criteria.

## Discussion

The overall prevalence of depression and/or anxiety from the KDHS data was 3.84% with a 95% CI of (3.56,4.14). The prevalence of depression and/or anxiety conducted in this study is at a diagnostic level. The two most prevalent mental health conditions among the general population are depressive and anxiety disorders, and because of their terrible effects, there are major public health issues. The participants in this study were told by physicians they have depression and anxiety compared to other studies which were assessed by screening tools. This finding is in line with other studies conducted in Ethiopia 4.14% [33]. In other words, this finding is lower than studies conducted in Pakistan 27.4% [34], Nigeria 20.5% [35], and China 14.2% [36]. The main possible reason for this discrepancy could be the different measurement tools for instance Zung's depression and anxiety self-rating scale screening tool was used but, our study was based on their diagnosis level told by the physician [36]. The other reason for this disparity can result from the sociocultural differences between the study participants.

Regarding factors associated with depression and/or anxiety was being sexually violated. This finding was concordant with other studies conducted in other countries United States [37,38] The probable reason for this association might be the effect of the emotional trauma of being violated that results in depression and or anxiety [37]. The other probable evidence for this association could be the impact of stigma from the other individual that leads to self-isolation and finally makes them anxious and/or depressed.

The other factor associated with depression and/or anxiety was having a chronic medical illness. This association is consistent with former studies conducted in the United States [39,40]. The probable reason for the association might be explained by dysregulation of specific Hypothalamic-pituitary-adrenocortical biological mechanisms, such as hemostasis of sympathetic nerve systems, which contribute to the pathophysiology of both physical and mental disorders [39]. The other reason for this association could be the disadvantages both at work and in private life that come from being unable to earn money for their needs [40]. There might be a bidirectional association that cannot be checked by a cross-sectional study physical illness can lead to mental disorders and vice versa.

Being divorced and widowed was the factor associated with depression and/or anxiety in Malaysia [17] and Ethiopia [33] likewise, this factor was also associated significantly. The reason for the associations could be because being divorced/ widowed had a psychological impact of loneliness isolation disconnect from social support [17]. Another evidence might be the

effect of dependence that comes from maltreatment by others because no one has belonged to them for protection. Having a job was another factor associated with depression and/or anxiety. This is in line with former studies conducted in Ethiopia [33]. The possible reason for the association could be justified as people's employment demands surpass their capacity for coping, which could lead to more stress risk of depression symptoms developing [33]. The other reason for the associations could be the effect of job-related stress which can persist for a prolonged time and lead to depression and/or anxiety.

Another factor that was associated with depression and/or anxiety was being HIV positive. This association was consistence with previous studies conducted in Ethiopia [41,42]. The possible reason for the association could be due to the psychological effect of receiving the diagnosis of HIV-positive, the stigma and emotional fallout might trigger a depressive and anxiety episode or relapse of the disorders [42]. The other possible evidence for the association could be the impact of HIV on losing their work, fear of being disregarded by others, and consequently difficulty in financial troubles [41]. The other evidence for this association might be the effect of the HAART drug and its side effects on the brain, the virus by itself, and the psychological effect of stigma/discrimination. Depression and anxiety are the most prevalent disorders with high public health burdens. This national finding could provide a clue for policymakers to mitigate depression and anxiety disorders. All concerned bodies; the Kenya National Health Office and other stakeholders are advised to mitigate the burden of depression and anxiety more than the effort used.

## Limitations of the study

Although this study has many strengths, it also has limitations inherent to the cross-sectional study design. The other weakness of this study is recall bias and social desirability bias to inform they have depression and or anxiety.

## Conclusions

The finding of this study revealed the actual diagnosis of depression and/or anxiety which is 3.84%. Sexually violated, having a chronic medical illness, being divorced and widowed, having a job, and being HIV positive were associated with depression and/or anxiety. The impact of depression and anxiety is a great health challenge that needs fast interventions to mitigate its global burden in Kenya. The Kenya policymakers and stakeholders are expected to reduce the risk factors of depression and/or anxiety especially sexual violations and HIV ADIS distributions.

## Acknowledgment

We would like to ensure the MEASUR DHS who allowed to access this dataset to conduct this secondary data analysis.

## Author contributions

**Conceptualization:** Mamaru Amsalu.

**Formal analysis:** Setegn Fentahun.

**Funding acquisition:** Girmaw Medfu Takelle.

**Project administration:** Gebresilassie Tadesse.

**Software:** Gidey Rtbey.

**Supervision:** Girum Nakie.

**Validation:** Fantahun Andualem, Yilkal Abebaw Wassie.

**Writing – original draft:** Mamaru Amsalu.

**Writing – review & editing:** Mamaru Amsalu.

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
