## [Decision Letter · Decision Letter 0]

26 Dec 2024

PONE-D-24-00891Determinants of depression and/or anxiety among the adult community in Kenya: Multilevel analysis of Kenya Demographic Health Survey of 2022PLOS ONE

Dear Dr.Mamaru Melkam Amsalu,

Thank you for submitting your manuscript to PLOS ONE. After careful consideration, we feel that it has merit but does not fully meet PLOS ONE’s publication criteria as it currently stands. Therefore, we invite you to submit a revised version of the manuscript that addresses the points raised during the review process.

Please submit your revised manuscript by  Feb 09 2025 11:59PM, If you will need more time than this to complete your revisions, please reply to this message or contact the journal office at plosone@plos.org . Please include the following items when submitting your revised manuscript:

We look forward to receiving your revised manuscript.

Kind regards,

Kamalakar Surineni, MD, MPH

Guest Editor

PLOS ONE

Journal Requirements:

Additional Editor Comments (if provided):

As noted by the reviewers, the paper needs considerable improvements but it still holds value. Please address and resubmit for further consideration.

Reviewers' comments:

Reviewer's Responses to Questions

**Comments to the Author**

1. Is the manuscript technically sound, and do the data support the conclusions?

Reviewer #1: No

Reviewer #2: No

Reviewer #3: Yes

2. Has the statistical analysis been performed appropriately and rigorously? 

Reviewer #1: No

Reviewer #2: I Don't Know

Reviewer #3: I Don't Know

3. Have the authors made all data underlying the findings in their manuscript fully available?

Reviewer #1: Yes

Reviewer #2: Yes

Reviewer #3: Yes

4. Is the manuscript presented in an intelligible fashion and written in standard English?

Reviewer #1: No

Reviewer #2: No

Reviewer #3: No

5. Review Comments to the Author

Reviewer #1: As the reviewer, I have conducted a detailed assessment of the manuscript titled "Determinants of depression and/or anxiety among the adult community in Kenya: Multilevel analysis of Kenya Demographic Health Survey of 2022" using the STROBE Statement checklist for observational studies. Here is an in-depth report addressing each of the checklist items and providing specific line references where issues have been identified:

Title and Abstract:

1. (a) The title accurately indicates the study's design as a multilevel analysis of Kenya demographic health survey of 2022.

• But it was also better to put the specific statistical model like multilevel binary logistic regression analysis as a study design.

(b) The abstract did not effectively describe what was indicated the title.

• The study revealed the overall prevalence of depression and/or anxiety among the community in Kenya from Kenya Demographic and Health Survey data of 2022, but the title you indicated and the objective you described in the abstract are completely unrelated. So, how do you relate the title with that of line 29-30?

• Line 32-33 needs to be rewrite.

• Line 33-34 is better suited for only multilevel analysis, as it’s reasoned for both Multilevel and logistic regression is inappropriate. Because the clustering nature of the data is not a reason for both multilevel and binary logistic regression analysis.

• Line 35 “The overall prevalence of 3.84% with a 95% CI of (3.56, 4.14)”, Whose prevalence, is it? It needs to be corrected.

• Line 36 “In multivariant multilevel logistic regression”, what does that mean multivariant? How do you relate with the word “multivariate” and “multivariable”.

• The keywords (line 45) you indicated is not enough.

• Overall, the abstract is not effectively described what was done.

Introduction:

2. The introduction section presents several issues:

• Line 54 and line 60: DSM-5 and WHO need to be abbreviated at first.

• Line 91-98: font size is not similar with others.

• Line 97-98: “This study aims to determine the associated factors of depression and anxiety with their burden among adult communities in Kenya from Kenya Demographic and Health Survey data of 2022”, how do you relate with the indicated objective in the abstract part? Even with that of indicated title? What is the difference between “Determinants” and “associated factors”?

• The introduction lacks the prevalence of depression in Africa particularly in Keny and lacks relevant literature to establish the rationale.

• Overall, the introduction is poorly written.

Objectives:

3. The manuscript does not clearly state specific study objectives, which is a critical omission. Specific objectives should be explicitly defined, including any prespecified hypotheses.

Methods:

4. The manuscript accurately presents the key elements of the study design, appropriately considering it as a secondary data analysis.

5. The description of the setting, locations, and data collection dates is correctly provided.

6. Since this is a secondary data analysis, the absence of eligibility criteria and participant selection details is acceptable so I didn’t see such thing.

7. The dependent and independent variables are not well-defined. How do you measure your dependent variable? Even if, the dependent variable was used from the KDHS definitions as reported in 2022, you should have established its measurement with relevant literature or you can operationalize it as well. Mention its indictor.

8. While all independent variables of the study indicated, sources of data for each variable and their method of data collection are not mentioned. You should have to describe each independent variable.

9. Why Stata version 14? Why not 17?

10. The manuscript explains how quantitative variables were handled in the analyses, including the use of different analytical techniques. However, irrelevant sentences types are described (Lines 126-127), which should be removed.

11. The statistical methods, including multilevel logistic regression to control for potential confounding variables, are well-described. However, it is not indicated how missing data were analyzed.

Results:

12. There are several issues in the results section:

• Line 170:” deviation” how do you relate with that of deviance?

• The reference error "Error! Reference source not found" needs clarification.

• Missing data in baseline variables should be explained, along with the methods used to address the missing data.

• The manuscript should clarify the model fit statistics referenced in the text (Line 376). It is not clear how someone knows whether the given result is COR or AOR.

• While the third model contained community-level variables (Residence: lines 135-136), it is not statistically significant at 5% level of significance as presented in Table 3. So, do you think that multilevel analysis is suitable or appropriate statistical method for the given data set? Is it relevance?

Discussion:

13. The 'Discussion' section can be generally divided into 3 separate paragraphs. 1) Introductory paragraph/rationale of the study, 2) Intermediate paragraphs/compare and contrast with the most recent and relevant literature, 3) Concluding paragraph/indicating future directions. The introductory paragraph contains the main idea of performing the study question.

• While the manuscript mentions the large sample size as a strength indicating there is adequate power to detect the true effect of the independent variable. However, it does not provide a power calculation to support this claim.

• Some limitations are discussed, but the limitation related to the 24-hour recall for bias is not mentioned and there is potential social desirability bias that could impact on the results. Discuss finding should be interpreted with caution.

Recommendation

This comprehensive report outlines the issues identified throughout the manuscript, with specific line references and requires substantial major revisions to address these concerns and enhance the clarity, relevance, and presentation of results. After making the necessary major revisions with re-analysis of the data, a re-evaluation is recommended for considering publication.

Reviewer #2: Overall, the premise and intention of the research by the authors is notable. However, the research question appears to be limited. The paper only calculates the percentage of individuals who received a diagnosis of depression and anxiety by a physician, and this ultimately is too limited to be generalized as prevalence of these conditions in the general population. Taking this alone does not adequately answer the question of general prevalence in Kenya.

The research question also appears to be simply a calculation from a dataset. It would be beneficial to have a more involved research question that takes into account other factors either in addition to prevalence or factors influencing prevalence. Other potential factors that are listed (such as being sexually violated) appear to be seen by chance rather than postulated or hypothesized in the beginning. There does not appear to be a clear hypothesis from the authors in the beginning and this can make the findings appear random and incohesive in nature. The limitations of the study were also not clarified adequately. Lastly, there appears to be a lack of clarity in the English grammar at times.

Although the intention of the authors is commendable, and this is a good topic for research, the overall methodology and research question requires significant revision.

Reviewer #3: The study presents an insightful multilevel analysis of the determinants of depression and anxiety in Kenya using the Kenya Demographic Health Survey 2022 data. The strengths of the study are:

1. Addressing mental health issues in Kenya is significant, considering the limited number of studies in this area.

2. Using data from a nationally representative survey enhances the validity and generalizability of findings.

3. The use of multilevel logistic regression considers the clustering nature of the data, which is methodologically appropriate.

4. The study identifies critical determinants like sexual violence, chronic illness, HIV status, and marital status, which can guide interventions.

I have the following recommendations:

1. There are many grammatical errors and some sentences are disjointed. Would advise a recheck of the manuscript or use assistance of writing services.

2. The study design and data extraction methods are described but lack clarity regarding the tools used for measuring depression and anxiety. Provide details on how depression and anxiety were diagnosed (e.g., DSM criteria, specific screening tools).

3. The reported prevalence of 3.84% seems low compared to global studies, potentially underestimating the problem. Discuss the implications of relying on physician-diagnosed cases rather than standardized screening tools.

4. While statistical methods are robust, the discussion of model fit (AIC, BIC) and ICC is limited. Elaborate on these metrics to justify the selection of the final model.

5. The discussion relies heavily on previous studies without critically analyzing the study's unique findings. Expand the discussion to include potential cultural or systemic explanations for the identified associations.

6. Recommendations for interventions are broad and not Kenya-specific. Propose tailored policy measures considering Kenya's healthcare infrastructure and cultural context.

7. Reorganize the introduction to clearly define the study's objectives and significance.

6. PLOS authors have the option to publish the peer review history of their article (what does this mean? ). If published, this will include your full peer review and any attached files.

**Do you want your identity to be public for this peer review?** For information about this choice, including consent withdrawal, please see our Privacy Policy .

Reviewer #1: No

Reviewer #2: No

Reviewer #3: **Yes: ** Nikhil Tondehal

---

## [Author Response · Author response to Decision Letter 1]

8 Jan 2025

Cover letter

Revision and resubmission of the manuscript with ID Number PONE-D-24-00891. Before all, I would like to thank the editorial teams on behalf of the authors, Plos One Journal regarding the fast review procedure of the manuscript titled “Prevalence and determinants of depression and/or anxiety among adults using Kenya Demographic and Health Survey of 2022: Multilevel logistic regression analysis” we thank you, for all the time and energy you devoted and the reviewers invested in offering feedback on our article, as well as for your constructive comments and suggestions. Lastly, we would like to confirm that this paper has not already been published or is not being considered by another journal for publication. All authors have approved the manuscript and agreed with its resubmission to Plos One Journal.

With regards!

On behalf of all the co-authors,

Mamaru Melkam, Correspondence author.

The authors have declared that no competing interests exist.

The authors received no specific funding for this work.

Response to the editor’s

We sincerely appreciate your constructive comments and suggestions for further improving our article. Based on your feedback and recommendations, we have made corrections and changes to the manuscript revising the whole part. Please find the point-by-point response to the reviewers' comments and recommendations below in blue color. The manuscript is amended in track change in a red color. Additionally, efforts were made to improve the language or typo errors to be simple for the understanding of readers.

Response to the reviewer's comments

On behalf of the authors, I'd like to express my gratitude to the editorial board, academic editor, and reviewers for your insightful comments that helped to increase the quality of the manuscript to become more scientifically sound. We thank all of the dear reviewers and editors for the constrictive comment to make our manuscript more improved. All the concerns raised by the reviewers and editor were tried to be addressed and the suggestions were also accepted. Currently, the article has undergone significant advancement as a result of the suggestions made by editing teams and reviewers from the initial submission to the present time. As you suggested, we have uploaded the track change and the cleaned revised manuscript.

Reviewer comments

Reviewer 1

As the reviewer, I have conducted a detailed assessment of the manuscript titled "Determinants of depression and/or anxiety among the adult community in Kenya: Multilevel analysis of Kenya Demographic Health Survey of 2022" using the STROBE Statement checklist for observational studies. Here is an in-depth report addressing each of the checklist items and providing specific line references where issues have been identified:

Response: Thank you very much for the comments and suggestions you provided to enhance the scientific quality of this manuscript to be easily understandable for the readers.

Title and Abstract:

1. (a) The title accurately indicates the study's design as a multilevel analysis of the kenya demographic health survey of 2022.

• But it was also better to put the specific statistical model like multilevel binary logistic regression analysis as a study design.

Response: Thank you for the comment; we have incorporated as you recommended.

(b) The abstract did not effectively describe what was indicated the title.

• The study revealed the overall prevalence of depression and/or anxiety among the community in Kenya from Kenya Demographic and Health Survey data of 2022, but the title you indicated and the objective you described in the abstract are completely unrelated. So, how do you relate the title with that of line 29-30?

Response: Thank you for the suggestion; we amended the abstract introduction part to be consistence with the title as you suggested.

• Line 32-33 needs to be rewrite.

Response: Thank you for the concerns; it is rewritten based on your recommendations.

• Line 33-34 is better suited for only multilevel analysis, as it’s reasoned for both Multilevel and logistic regression is inappropriate. Because the clustering nature of the data is not a reason for both multilevel and binary logistic regression analysis. Response: Thank you for the comment; it is refined as you suggested

• Line 35 “The overall prevalence of 3.84% with a 95% CI of (3.56, 4.14)”, Whose prevalence, is it? It needs to be corrected.

Response: Thank you for the suggestion; we made it the prevalence of depression and/or anxiety.

• Line 36 “In multivariant multilevel logistic regression”, what does that mean multivariant? How do you relate with the word “multivariate” and “multivariable”.

Response: Thank you for the comment; we have amended it as multivariable. The multivariate methods are not the same as multivariable methods. Multivariate methods have more than one dependent variable or place variables on an equal footing. Multivariable methods have one dependent variable and more than one independent variable or covariates.

• The keywords (line 45) you indicated is not enough.

Response: Thank you for the concern; we add more words.

• Overall, the abstract is not effectively described what was done.

Response: Thank you for the comment and suggestions we have made so much improvement on the abstract.

Introduction:

2. The introduction section presents several issues:

• Line 54 and line 60: DSM-5 and WHO need to be abbreviated at first.

Response: Thank you for the comment; they are correct as you suggested.

• Line 91-98: font size is not similar with others.

Response: Thank you for the suggestion; it is amended.

• Line 97-98: “This study aims to determine the associated factors of depression and anxiety with their burden among adult communities in Kenya from Kenya Demographic and Health Survey data of 2022”, how do you relate with the indicated objective in the abstract part? Even with that of indicated title? What is the difference between “Determinants” and “associated factors”?

Response: Thank you for the comment; we made the last paragraph of the introduction consistence with the abstract and title by making determinants.

• The introduction lacks the prevalence of depression in Africa particularly in Keny and lacks relevant literature to establish the rationale.

Response: Thank you for the concern; we have included more prevalence as you recommended.

• Overall, the introduction is poorly written.

Response: Thank you for the comment; we improved the entire introduction much better based on your comments.

Objectives:

3. The manuscript does not clearly state s/pecific study objectives, which is a critical omission. Specific objectives should be explicitly defined, including any prespecified hypotheses.

Response: Thank you for the comment; the aims or objectives of this study were mentioned in the abstract and introduction last paragraph, please take a look at manuscript line number (117,123).

Methods:

4. The manuscript accurately presents the key elements of the study design, appropriately considering it as a secondary data analysis.

5. The description of the setting, locations, and data collection dates is correctly provided.

6. Since this is a secondary data analysis, the absence of eligibility criteria and participant selection details is acceptable so I didn’t see such thing.

Response: Thank you for the comment; the eligibility criteria were the study participants who were not avail at home during the data collection period for this DHS data collection.

7. The dependent and independent variables are not well-defined. How do you measure your dependent variable? Even if, the dependent variable was used from the KDHS definitions as reported in 2022, you should have established its measurement with relevant literature or you can operationalize it as well. Mention its indictor.

Response: Thank you for the suggestion; DHS measures depression and anxiety based on the diagnostic criteria. Study participants leveled as they have depression and anxiety by physicians.

8. While all independent variables of the study indicated, sources of data for each variables and their method of data collection are not mentioned. You should have to describe each independent variable.

Response: Thank you for the comment; the data were collected through the interview-based survey method. The data were collected interview-based data collection for all independent variables.

9. Why Stata version 14? Why not 17?

Response: Thank you for the concern; because of we can’t access the updated version 17 for this analysis.

10. The manuscript explains how quantitative variables were handled in the analyses, including the use of different analytical techniques. However, irrelevant sentences types are described (Lines 126-127), which should be removed.

Response: Thank you for the suggestion; we have amended based on your suggestion.

11. The statistical methods, including multilevel logistic regression to control for potential confounding variables, are well-described. However, it is not indicated how missing data were analysed.

Response: Thank you for the comment; Missing values for the dependent variable were removed from this study. For the independent variables, the imputations method can be used for the missed data for determinant factors.

Results:

12. There are several issues in the results section:

• Line 170:” deviation” how do you relate with that of deviance?

Response: Thank you for the comment; we have conducted a deviance measurement among models which means deviation among the null model with each model. We have selected the best model with the lowest deviance.

• The reference error "Error! Reference source not found" needs clarification.

Response: Thank you for the concern; it is amended as you suggested.

• Missing data in baseline variables should be explained, along with the methods used to address the missing data.

Response: Thank you for the comment; the management of the missed value for the independent variables can be managed through the imputations method.

• The manuscript should clarify the model fit statistics referenced in the text (Line 376). It is not clear how someone knows whether the given result is COR or AOR.

Response: Thank you for the comment; we have used multivariable analysis which means all factors associated with the binary analysis with the outcome variables that indicate the odd ratio is AOR.

• While the third model contained community-level variables (Residence: lines 135-136), it is not statistically significant at 5% level of significance as presented in Table 3. So, do you think that multilevel analysis is suitable or appropriate statistical method for the given data set? Is it relevance?

Response: Thank you for the comment; we have included other community-level variable mass media exposure but both are not associated. We conducted this study by considering community-level variables by hypothesizing they might be important with a huge sample size analysis but the community variable may not be necessarily associated.

Discussion:

13. The 'Discussion' section can be generally divided into 3 separate paragraphs. 1) Introductory paragraph/rationale of the study, 2) Intermediate paragraphs/compare and contrast with the most recent and relevant literature, 3) Concluding paragraph/indicating future directions. The introductory paragraph contains the main idea of performing the study question.

Response: Thank you for the comment; we have tried to incorporate the points you have raised in the discussion including the rationale, compression, and conclusion.

• While the manuscript mentions the large sample size as a strength indicating there is adequate power to detect the true effect of the independent variable. However, it does not provide a power calculation to support this claim.

Response: Thank you for the comment; the calculated sample size for this topic is under 400 which is not comparable but we omit the strength from this study since strength is not that much mandatory.

• Some limitations are discussed, but the limitation related to the 24-hour recall for bias is not mentioned and there is potential social desirability bias that could impact on the results. Discuss finding should be interpreted with caution.

Response: Thank you for the suggestion; it is improved based on your suggestions.

Recommendation

This comprehensive report outlines the issues identified throughout the manuscript, with specific line references and requires substantial major revisions to address these concerns and enhance the clarity, relevance, and presentation of results. After making the necessary major revisions with re-analysis of the data, a re-evaluation is recommended for considering publication.

Response: Thank you for the comment and suggestion; we improved the entire manuscript to enhance the quality and to make it clear.

Reviewer #2: Overall, the premise and intention of the research by the authors is notable. However, the research question appears to be limited. The paper only calculates the percentage of individuals who received a diagnosis of depression and anxiety by a physician, and this ultimately is too limited to be generalized as prevalence of these conditions in the general population. Taking this alone does not adequately answer the question of general prevalence in Kenya. The research question also appears to be simply a calculation from a dataset. It would be beneficial to have a more involved research question that takes into account other factors either in addition to prevalence or factors influencing prevalence.

Response: Thank you for the comment; we have used the diagnosis criteria to get severe depression and anxiety since there are many studies conducted with screening tools in the world. This study provides a nationwide real diagnosis with DSM-V which is novel and not well studied so far.

Other potential factors that are listed (such as being sexually violated) appear to be seen by chance rather than postulated or hypothesized in the beginning. There does not appear to be a clear hypothesis from the authors in the beginning and this can make the findings appear random and incohesive in nature.

Response: Thank you for the concern; sexual violence has been associated among many studies with depression and anxiety. It was not random we have conducted this study with the stated aim of a national dataset of DHS; the national data is intentionally collected for such kind of study as far as we know.

The limitations of the study were also not clarified adequately. Lastly, there appears to be a lack of clarity in the English grammar at times.

Response: Thank you for the suggestion; we have improved based on your suggestion.

Although the intention of the authors is commendable, and this is a good topic for research, the overall methodology and research question requires significant revision.

Response: Thank you for the suggestion and comments you provided; we have improved this manuscript a lot to be clear for the readers.

Reviewer #3: The study presents an insightful multilevel analysis of the determinants of depression and anxiety in Kenya using the Kenya Demographic Health Survey 2022 data. The strengths of the study are:

1. Addressing mental health issues in Kenya is significant, considering the limited number of studies in this area.

2. Using data from a nationally representative survey enhances the validity and generalizability of findings.

3. The use of multilevel logistic regression considers the clustering nature of the data, which is methodologically appropriate.

4. The study identifies critical determinants like sexual violence, chronic illness, HIV status, and marital status, which can guide interventions.

Response: Thank you for the comments and suggestions that you raised which are really important to enhance the scientific quality of this manuscript to be easily understandable for the readers.

I have the following recommendations:

1. There are many grammatical errors and some sentences are disjointed. Would advise a recheck of the manuscript or use assistance of writing services.

Response: Thank you for the comment; the typo and grammatical errors were checked by English language and mental health experts

---

## [Decision Letter · Decision Letter 1]

5 Feb 2025

Prevalence and determinants of depression and/or anxiety among adults using Kenya Demographic and Health Survey of 2022: Multilevel logistic regression analysis

PONE-D-24-00891R1

Dear Dr. Mamaru Melkam Amsalu, 

We’re pleased to inform you that your manuscript has been judged scientifically suitable for publication and will be formally accepted for publication once it meets all outstanding technical requirements.

Kind regards,

Kamalakar Surineni, MD, MPH

Guest Editor

PLOS ONE

Additional Editor Comments (optional):

Thank you so much for improving the manuscript based on the reviewer feedback. I'm pleased to share that the manuscript now meets the publication criteria, and I’m happy to announce that it has been accepted!

Reviewers' comments:

Reviewer's Responses to Questions

**Comments to the Author**

1. If the authors have adequately addressed your comments raised in a previous round of review and you feel that this manuscript is now acceptable for publication, you may indicate that here to bypass the “Comments to the Author” section, enter your conflict of interest statement in the “Confidential to Editor” section, and submit your "Accept" recommendation.

Reviewer #1: All comments have been addressed

Reviewer #3: All comments have been addressed

2. Is the manuscript technically sound, and do the data support the conclusions?

Reviewer #1: Yes

Reviewer #3: Yes

3. Has the statistical analysis been performed appropriately and rigorously? 

Reviewer #1: Yes

Reviewer #3: Yes

4. Have the authors made all data underlying the findings in their manuscript fully available?

Reviewer #1: Yes

Reviewer #3: Yes

5. Is the manuscript presented in an intelligible fashion and written in standard English?

Reviewer #1: Yes

Reviewer #3: Yes

6. Review Comments to the Author

Reviewer #1: (No Response)

Reviewer #3: (No Response)

7. PLOS authors have the option to publish the peer review history of their article (what does this mean? ). If published, this will include your full peer review and any attached files.

**Do you want your identity to be public for this peer review?** For information about this choice, including consent withdrawal, please see our Privacy Policy .

Reviewer #1: **Yes: ** Abdu Hailu Shibeshi

Reviewer #3: **Yes: ** Nikhil Tondehal

---

## [Editor Report · Acceptance letter]

PONE-D-24-00891R1

PLOS ONE

Dear Dr. Amsalu,

I'm pleased to inform you that your manuscript has been deemed suitable for publication in PLOS ONE. Congratulations! Your manuscript is now being handed over to our production team.

Kind regards,

on behalf of

Dr. Kamalakar Surineni

Guest Editor

PLOS ONE